# Safety and Efficacy of Medical Cannabis in Fibromyalgia

**DOI:** 10.3390/jcm8060807

**Published:** 2019-06-05

**Authors:** Iftach Sagy, Lihi Bar-Lev Schleider, Mahmoud Abu-Shakra, Victor Novack

**Affiliations:** 1Department of Rheumatology, Rabin Medical Center, Petach Tikva 49100, Israel; iftachsagy@gmail.com; 2Cannabis Clinical Research Institute, Soroka University Medical Center and Faculty of Health Sciences, Ben-Gurion University of the Negev, Be’er-Sheva 84101, Israel; lihibarlev@gmail.com; 3Research Department, Tikun Olam LTD, Tel-Aviv 6296602, Israel; 4Department of Rheumatology, Soroka University Medical Center and Faculty of Health Sciences, Ben-Gurion University of the Negev, Be’er-Sheva 84101, Israel; mahmoud@bgu.ac.il

**Keywords:** medical cannabis, fibromyalgia, quality of life, chronic pain

## Abstract

Background: Chronic pain may be treated by medical cannabis. Yet, there is scarce evidence to support the role of medical cannabis in the treatment of fibromyalgia. The aim of the study was to investigate the characteristics, safety, and effectiveness of medical cannabis therapy for fibromyalgia. Methods: A prospective observational study with six months follow-up period based on fibromyalgia patients who were willing to answer questionnaire in a specialized medical cannabis clinic between 2015 and 2017. Results: Among the 367 fibromyalgia patients, the mean age was 52.9 ± 15.1, of whom 301 (82.0%) were women. Twenty eight patients (7.6%) stopped the treatment prior to the six months follow-up. The six months response rate was 70.8%. Pain intensity (scale 0–10) reduced from a median of 9.0 at baseline to 5.0 (*p* < 0.001), and 194 patients (81.1%) achieved treatment response. In a multivariate analysis, age above 60 years (odds ratio [OR] 0.34, 95% C.I 0.16–0.72), concerns about cannabis treatment (OR 0.36, 95% C.I 0.16–0.80), spasticity (OR 2.26, 95% C.I 1.08–4.72), and previous use of cannabis (OR 2.46 95% C.I 1.06–5.74) were associated with treatment outcome. The most common adverse effects were mild and included dizziness (7.9%), dry mouth (6.7%), and gastrointestinal symptoms (5.4%). Conclusion: Medical cannabis appears to be a safe and effective alternative for the treatment of fibromyalgia symptoms. Standardization of treatment compounds and regimens are required.

## 1. Introduction

Fibromyalgia is a common syndrome of chronic pain, often accompanied by sleeping disturbances, cognitive impairment, and psychiatric and somatic symptoms [1,2]. The prevalence of fibromyalgia is 2–8% of the entire population, and it is the most common reason for generalized pain among working age women worldwide [3,4].

Therapy for fibromyalgia is challenging and based on a multidisciplinary approach. Patients with fibromyalgia may respond to a combination of pharmacological (e.g., tricyclic antidepressants, serotonin/norepinephrine reuptake inhibitors, and anticonvulsants) and non-pharmacological interventions (e.g., aerobic exercise, cognitive-behavioral therapy, and rehabilitation programs) [5]. On the other hand, utilization of opioids was found to be associated with poorer symptoms and poorer functional and occupational status compared to nonusers [6].

Medical cannabis represents a promising therapeutic option for fibromyalgia patients due to its effectiveness and relatively low rate of serious adverse effects [7,8]. Although the identification of cannabinoid receptors and their endogenous ligands has triggered a large body of studies, there is a paucity of large-scale and prospective clinical trials regarding their role in fibromyalgia [9]. Only a handful of studies have examined the effect of medical cannabis on fibromyalgia. These studies had rather small sample sizes (31–40 subjects) and a short duration of follow up, which makes the generalizability of the results questionable [10,11,12]. In the current analysis of the prospective registry, we aim to investigate the safety and effectiveness of fibromyalgia patients receiving medical cannabis.

## 2. Experimental Section

### 2.1. Study Population

In Israel, patients prescribed medical cannabis are required to receive an approval from the Israel Medical Cannabis Agency (IMCA), a department within the Israeli Ministry of Health. Currently, there are more than 30,000 patients approved for medical cannabis use in Israel. Following the authorization, patients are asked to contact one of eight specified medical cannabis providers. Patients receive structured guidance by a certified nurse in the cannabis field regarding the available strains and route of administration. The monthly dose is set by the IMCA authorization according to the clinical indication. The patient then starts gradual titration process after choosing a strain according to his/her own decision Tikun-Olam Ltd. (TO) is the largest medical cannabis provider in Israel, which serves annually a third of the entire medical cannabis users in Israel.

This analysis of the prospectively collected data included all patients with diagnosis of fibromyalgia (primary or secondary to other conditions) who initiated treatment with medical cannabis in TO from January 2015 to December 2017. The data were extracted and analyzed retrospectively. The fibromyalgia diagnosis was established by a board-certified rheumatologist according to the American College of Rheumatology preliminary diagnostic criteria for fibromyalgia [13]. Patients were referred to cannabis treatment by ether the family physician, pain physician, or specialized rheumatologist after receiving treatment for at least a year without improvement. The study was approved by the Soroka University Medical Center (SUMC) institutional ethics committee and was conducted by the SUMC Clinical Cannabis Research Institute.

### 2.2. Data Collection

The intake questionnaire included demographic details, daily habits, substance abuse, medical background, concurrent use of other medications, symptoms checklist, and quality of life (QOL) assessment, stratified by components in 5 points Likert scale (e.g., sleep; appetite; sexual activity; and how a patient would assess their quality of life on a 5 points scale, with 1 being very poor and 5 being very good). Fibromyalgia symptoms after six months were assessed using 8-points Likert scale (1—severe symptomatic deterioration, to 8—maximal symptomatic improvement). A certified nurse educated the patients on the use of medical cannabis; gave instructions on route of administration according to the medical cannabis license (oil vs. inflorescence), delivery methods (drops, flowers, capsules, or cigarettes), and possible adverse effects; and provided an explanation on regulatory issues. The nurse also advised on selecting the cannabis strain (out of 14 strains available) and treatment dose according to titration protocol.

Cannabis products are composed of two major active components: tetrahydrocannabinol (THC) and cannabidiol (CBD). THC is the psychoactive component, which affects pain, appetite, orientation, and emotions, through CB1 and CB2 receptors. CBD has analgesic, anti-inflammatory, and anti-anxiety effects via a complex mechanism acting as a negative allosteric modulator of CB1 receptor [14]. The relative proportion of THC:CBD determines each strain’s type of effect, pharmacokinetics, and adverse events. In addition, more than 60 other cannabinoids have been identified, with a variety of clinical effects (e.g., anti-inflammatory and analgesic effects) and pharmacokinetics.

In this study, we used a gradual titration process rather than a fixed dose. Initially, all patients received a low dose of cannabis below the therapeutic effect (e.g., a drop of 15% THC-rich cannabis TID). Patients then were instructed to increase the dosage gradually in small intervals (e.g., a single drop per day) until they reached a therapeutic effect (e.g., subjective relief of their pain, significant improvement in their quality of life). In case of inflorescence (each cigarette contained 0.75 g of cannabis), patients were instructed to use one breath every 3–4 h, and to increase the amount gradually in this interval until therapeutic effect is reached. Mixing of oil and inflorescence at the same usage was not recommended. In case of adverse events, patients were instructed to use the last dosage that did not cause undesirable symptoms. The titration was similar for both THC- and CBD-rich strains. In addition, the cannabis provider operated a 24/7 call center to address any concerns that might have been raised by the patients. The final dosage depended on the primary indication for cannabis use, age, medical background, parallel use of other analgesic regimes, and previous exposure to cannabis. All patients underwent one and six month follow-up telephonic interviews. The later was extensive and included an assessment of the change in medical cannabis dose and regimen, change in QOL, disease- and medical cannabis-related symptoms, and alteration in the use and dosage of other medications.

### 2.3. Study Outcomes

For safety analysis, we assessed the frequency of medical cannabis-related side effects, including those of patients who ceased cannabis use before six months had passed. We also assessed patients’ perceptions regarding the change in fibromyalgia symptoms in the 6 month follow-up. The following symptoms were included: headaches, dizziness, nausea, vomiting, constipation, drop in sugar, drowsiness, weakness, dry mouth, cough, increased/lack of appetite, hyperactivity, restlessness, cognitive impairment, depression, anxiety, confusion, and disorientation. For disease-related symptoms, patients were asked to report whether each symptom disappeared, improved, deteriorated, or remained unchanged at six months follow up.

For effectiveness analysis, the primary outcome was treatment response, defined as at least moderate or significant improvement in a patient’s condition at six months follow-up without the cessation of treatment or serious side effects. Patients lost to follow-up were considered as failures for the purposes of the effectiveness analysis. In addition, we assessed the following secondary outcomes:Pain intensity—assessment by the numeric rating scale (NRS) with an 11-point scale (0 = no pain, 10 = worst pain imaginable).Quality of life—global assessment by the patient using the Likert scale with five options: very good, good, neither good nor bad, bad, or very bad.Perception of the general effect of cannabis—global assessment by using the Likert scale with seven options: significant improvement, moderate improvement, slight improvement, no change, slight deterioration, moderate deterioration, or significant deterioration.

### 2.4. Statistical Analysis

Continuous variables with normal distribution were presented as means with standard deviation. Ordinal variables or continuous variables with non-normal distribution were presented as medians with an interquartile range (IQR). Categorical variables were presented as counts and percent of the total. We used t-test for the analysis of the continuous variables with normal distribution. The non-parametric Wilcoxon test was used whenever parametric assumptions could not be satisfied. We utilized logistic regression for the multivariate analysis of factors associated with treatment success to control possible confounders. The final model was selected according to the statistical significance of coefficients, their clinical relevance, and the model discriminatory characteristic, which were evaluated by calculating the c-statistic, in addition to choosing the minimal −2 log likelihood of each model. We considered a *p*-value of 0.05 or less (two-sided) as statistically significant. IBM SPSS software, version 25.0, was used for statistical analysis.

## 3. Results

### 3.1. Cohort Characteristics

We identified 367 patients with fibromyalgia who had started the treatment with medical cannabis. During the study period, 35 received medical cannabis for less than six months and were not eligible for six months follow-up, 28 stopped medical cannabis treatment before six months follow-up, four switched to another medical cannabis supplier, and two died within the first six months (Figure 1). Out of the remaining 298 patients treated for six months, 211 responded with the follow-up questionnaire (70.8% response rate). In addition, out of the 87 patients who did not respond to the six months questionnaire, 76 patients (87.3%) were using cannabis at six months. To minimize selection bias, we compared baseline characteristics among six months respondents and non-respondents. As shown in Appendix A, there were no differences in baseline characteristics among those who responded to the six months follow-up questionnaire compared to those who did not.

Table 1 shows baseline characteristics of the study population. The majority of the patients were 40–60 years old (181 patients, 49.3%) and female (301 patients, 82.0%) with BMI of 28.6 ± 18.2 kg/m^2^. Patients had reported previous experience with recreational cannabis in the past in 45.2% of cases. The median length of fibromyalgia symptoms was 7 years, and 320 (87.2%) patients reported constant daily pain. In 283 patients (77.1%), fibromyalgia was the primary pain-related indication to initiate medical cannabis therapy. Fibromyalgia was the secondary indication to initiate cannabis therapy in 35 (9.5%) patients with cancer, 22 (6.0%) patients with post-traumatic stress disorder (PTSD), and in 27 (7.4%) patients with other indications.

The median cannabis approved dosage was 670 mg/day (inter-quartile range 670–670 mg) at initiation and 1000 mg/day (inter-quartile range 700–1000 mg) at six months (*p* = 0.01). The median THC and CBD dosages at six months were 140 mg/day (inter-quartile range 90–200 mg) and 39 mg/day (inter-quartile range 10–69 mg), respectively. When comparing dose at six months between patients with fibromyalgia as a primary or secondary indication, the primary fibromyalgia patients utilized the same THC dosages as the secondary patients (median of 140.0 mg/day for both, *p* = 0.95) and similar CBD dosages (median of 40.0 vs. 28.0 mg/day respectively, *p* = 0.52).

### 3.2. Safety Analysis

At treatment initiation, 328 (89.4%) patients received 20 g or less of cannabis per month, which was administrated to 247 (67.3%) patients using inflorescence (Table 1). During the study follow-up, a total mean of 3.3 regimens was prescribed per patient, with a total of 952 (56.4%) THC-rich regimens used compared to 129 (21.7%) CBD-rich regimes (Appendix A).

Medical cannabis-related adverse events, reported by patients six months after cannabis use, are shown in Appendix A. Overall the most common symptoms were dizziness reported by 19 patients (7.9%), dry mouth by 16 patients (6.7%), nausea/vomiting by 13 patients (5.4%), and hyperactivity by 12 patients (5.5%).

### 3.3. Effectiveness Analysis

The overall treatment success was achieved in 194 out of 239 patients (81.1%)—proportion of patients reporting at least moderate improvement in their condition while still receiving medical cannabis without experiencing serious adverse events out of patients who either responded to the six months questionnaire or stopped the treatment (Figure 2). Comparison of fibromyalgia-related symptoms among patients at intake and at six months follow-up is shown in Appendix A. The sleep problems reported by 196 patients (92.9%) at intake improved in 144 patients (73.4%) and disappeared in 26 patients (13.2%, *p* < 0.001). Depression-related symptoms reported by 125 patients (59.2%) at the baseline improved in 101 patients (80.8%, *p* < 0.001).

In a multivariate logistic regression (Table 2), age above 60 (O.R 0.34, 95% C.I 0.16–0.72) and concerns about cannabis treatment (O.R 0.36, 95% C.I 0.16–0.80) were associated with treatment failure, whereas spasticity at treatment initiation (O.R 2.26, 95% C.I 1.08–4.72) and previous use of cannabis (O.R 2.46 95% C.I 1.06–5.74) were associated with treatment success.

Figure 3 shows the evaluation of pain intensity (presented in NRS 11 points scale) at baseline and six months follow-up. Prior to treatment initiation, 193 patients (52.5%) reported a high level of pain scale (8–10). However, after six months of follow-up, only 19 patients (7.9%) reported similar pain intensity. Overall pain intensity reduced from a median of 9.0 (inter-quartile range 8.0–10.0) at baseline to 5.0 (inter-quartile range 4.0–6.0) after six months (*p* < 0.001).

The evaluation of QOL (in 5 points Likert scale) prior to and after six months of medical cannabis treatment is shown in Figure 4. Whereas prior to treatment initiation 10 patients (2.7%) reported good or very good QOL, after six months of treatment 148 patients (61.9%) reported their QOL to be good or very good (*p* < 0.001). When analyzing QOL components, sleep quality, appetite, and sexual activity significantly improved at six months (*p* < 0.001, 0.02, and 0.03 respectively, Appendix A). Other components (e.g., mobility, dressing, and concentration) did not improve, and the quality of daily activities deteriorated at six months (*p* < 0.001).

### 3.4. Additional Regimens

The change in the utilization of other drugs for the treatment of fibromyalgia after six months is shown in Appendix A. Most patients ceased, reduced, or at least did not change the dosage of their chronic drugs for fibromyalgia while receiving medical cannabis. At six months, 28 out of 126 patients (22.2%) stopped or reduced their dosage of opioids (<0.001), and 24 out of 118 (20.3%) reduced their dosage of benzodiazepines (*p* < 0.001). When stratifying the analysis to patients with primary vs. secondary fibromyalgia (Appendix A), both groups show the same improvement at six months in terms of pain intensity and overall quality of life.

## 4. Discussion

In the present study, we demonstrated that medical cannabis is an effective and safe option for the treatment of fibromyalgia patients’ symptoms. We found a significant improvement in pain intensity and significant improvement in patients’ overall quality of life and fibromyalgia-related symptoms after six months of medical cannabis therapy. In addition, there were relatively minor adverse effects with a small number of patients who discontinued the use at six months. To the best of our knowledge, this is the first trial to use herbal cannabis in fibromyalgia patients.

A search of the current literature has identified three randomized controlled trials evaluating the effect of medical cannabis on fibromyalgia-related symptoms. Skrabek et al. enrolled 32 patients to receive nabilone, an orally administered cannabinoid, vs. placebo therapy [10]. At four weeks follow-up there was a significant decrease of 2 points of NRS in addition to improvement in anxiety and overall quality of life. Ware et al. enrolled 29 patients in a trial of nabilone vs. amitriptyline to investigate the effect on sleep disorders among fibromyalgia patients over 2 weeks of therapy. The authors found a moderate effect on insomnia, but not on other aspects of sleep, in addition to no improvement in pain and quality of life [11]. Lastly, Fiz et al. enrolled 56 patients to receive either medical cannabis (the type is not mentioned) or standard therapy [15]. The authors reported a significant effect on pain two hours from consumption, with no effect on quality of life or sleep disorders. Data regarding pain intensity longer than 2 h were not available. Compared to the studies mentioned above, our study has several advantages. First, our study represents a real-world experience of herbal cannabis use in the cohort of patient with fibromyalgia. Second, we have assessed a substantially larger cohort of 367 fibromyalgia patients with six months follow-up of 211 patients (vs. 30–56 patients in previous studies). Our data also provided a relatively long follow-up of six months periods (compared to only several weeks follow up), which allowed us to analyze the effect and safety of medical cannabis on fibromyalgia patients over an extended period of time. Lastly, we studied the effect of medical cannabis on every aspect of fibromyalgia: improvement in chronic pain, quality of life, disease perception and specific symptoms, and the incidence of adverse effects.

There are several pharmacological regimes that are recommended to treat fibromyalgia [5]. However, their efficacy is relatively limited. The use of low-dose amitriptyline, a tricyclic antidepressant, was associated with 30% reduction in pain level with minor effect on sleep quality. A similar pain reduction rate was shown in meta-analyses of both anticonvulsants and serotonin–noradrenalin reuptake inhibitors [16,17]. However, withdrawal rates due to side effects in these studies were higher compared with placebo. Our results pointed out that cannabis may be at least equal to these regimes, yet with minor adverse effects that resulted in low dropout rates in our study.

Medical cannabis use was reported to be associated with a change in the utilization of other prescription regimens [18,19,20]. In our cohort, after six months of medical cannabis therapy, a substantial fraction of patients stopped or decreased the dosage of other medical therapies. Of note, 22.2% of opioids users at the baseline reduced or ceased the use of these medications at six months follow-up. Considering that opioid use is coupled with a complex titration process, higher risk for dependency, and a higher rate of serious adverse effects, medical cannabis may pose a reasonable therapeutic alternative [21,22,23].

Previous studies have shown that medical cannabis use was more prevalent among young adults and males [24,25]. However, our cohort was composed of a majority of 40–60 years old women, representing the population most affected by fibromyalgia [26,27]. These findings correlate with more recent reports that indicate a substantial increase in the age of medical cannabis users [28,29]. Although patients baseline NRS was considerably high (9/10), it represents patients who failed to respond to the standard therapy during a follow up of at least a year. Thus, our study cohort represents severe and poorly controlled fibromyalgia patients, which explains the higher symptomatic burden. Previous studies reported similar characteristics. For instance, Fiz et al. reported 37 mm VAS decrease two hours after cannabis administration (baseline VAS was 80mm) [15]. Goldenberg et al. reported a mean VAS of 81.5 mm among placebo users compared to fluoxentine- and amitriptyline-treated fibromyalgia patients [30].

Patients in our and other studies are often reporting that medical cannabis is more tolerable and with fewer adverse events compared to other therapies [31]. Similar to previous studies, we found that medical cannabis use is safe among fibromyalgia patients [7,32]. At six months follow-up, there was a relatively low rate of minor adverse events, and only 28 patients (7.6%) stopped using medical cannabis. In concordance with the literature, we found that dizziness, dry mouth, hyperactivity, drowsiness, and gastrointestinal symptoms are all possible adverse effects of cannabis use [14,33].

In our cohort, we had a relatively low rate of adverse events. For instance, the most commonly reported adverse events after six months were dizziness (7.9%), dry mouth (6.7%), and vomiting/nausea (5.4%). Yet, comparing our findings to other studies using the same titration approach yields similar rates of the adverse events. For instance, among 2736 elderly patients (65 and older) who used medical cannabis, dizziness was reported by 9.7% after six months of use [8]. First, as mentioned above, this may be associated with the gradual titration process, which may lead to the mitigation of most of cannabis’ adverse effects. Second, the evaluation of adverse events occurred only after six months of therapy. Since most of the patients developed tolerance to adverse effects in days, this may have led to lower rate of reported adverse events at six months follow-up. These findings further support the previously suggested cannabis titration approach of “start low, go slow, and stay low” to minimize both adverse events and the risk of addiction [14]. Lastly, the majority of our cohort used relatively low dosage (20 g or less per month) of cannabis at baseline and after six months (89.4% and 78.1%, respectively). The mean THC and CBD did not change between the first and last month of follow-up. These findings can also explain the low rate of adverse events, which were mostly dose-dependent. Clinicians should be aware of unjustified dose escalations (e.g., above 3 g/day in non-cancer patients) to prevent misuse or addiction to cannabis [34].

We found that patients’ concerns and worries regarding cannabis prior to treatment initiation were associated with lower odds of treatment success, whereas previous experience with cannabis was associated with treatment success. We acknowledge that these findings and the observational nature of our study could constitute evidence for the strong placebo effect associated with cannabis use, and emphasize the importance of double-blinded clinical trials, especially when testing subjective outcomes such as pain and quality of life. Yet, even blinded clinical trials may be biased towards overestimating the effectiveness of medical cannabis due to the lack of the psychoactive effect of placebo substances [35].

Our study has several important limitations. First, this study was of an observational nature and could not establish causality between medical cannabis use and improvement in fibromyalgia outcomes. The improvement at six months may be alternatively explained by regression to the mean phenomenon. Since this was not a randomized controlled trial, we can recommend neither a specific dosage nor specific cannabis product. Second, the close to 30% non-respondent rate in the six months follow-up may have resulted in a non-response bias. Yet, there were no significant differences between respondents to the non-respondents at the baseline characteristics, and more than 85% of the non-respondents were still using medical cannabis at six months. In addition, we cannot evaluate the actual compliance on a monthly basis. In concordance with the vast majority of studies, data on the actual utilization of cannabis were not available. Third, the fibromyalgia diagnosis was established by the referring rheumatologist; therefore, we could not verify that the American College of Rheumatology preliminary diagnostic criteria for fibromyalgia were fulfilled in every case [13]. Fourth, we had no control group to compare the clinical outcomes of medical cannabis use. Hence, some of the improvement may be attributed to spontaneous improvement in the course of the disease rather than medical cannabis utilization. Moreover, the patients in this study used 14 different strains, which prevented us from conducting a comparison between THC and CBD strains and products in terms of effectiveness and safety. Fifth, the change in the utilization of other drugs (than cannabis) for the treatment of fibromyalgia was based on self-reports and was prone to recall biases. Yet, we showed that most patients ceased, reduced, or at least did not change the dosage of their chronic drugs for fibromyalgia while receiving medical cannabis. Additionally, although we found that cannabis use is relatively safe among fibromyalgia patients, the conclusion should not be made about safety while driving under the influence of cannabis, as it was not a measured outcome of this study. The data of this study was provided by a registry that included cannabis users with several clinical indications. Hence, the questionnaire that was used did not include specific symptoms of fibromyalgia (e.g., fibro fog). Lastly, at this stage of medical cannabis research, we are not in a position to identify and thus synthesize single or multiple agents that are responsible for the therapeutic effects.

## 5. Conclusions

Notwithstanding these limitations, the present observational study innovates by showing that medical cannabis may be an effective and safe treatment to fibromyalgia in a large cohort with six months follow up. Our data indicates that medical cannabis could be a promising therapeutic option for the treatment of fibromyalgia, especially for those who failed on standard pharmacological therapies. We show that medical cannabis is effective and safe when titrated slowly and gradually. Considering the low rates of addiction and serious adverse effects (especially compared to opioids), cannabis therapy should be considered to ease the symptom burden among those fibromyalgia patients who are not responding to standard care. Moreover, our results highlight the need for further research to identify the effect of cannabis on other clinical conditions that are associated with fibromyalgia: cognitive impairment, fatigue, and additional chronic pain syndromes. Future studies should aim to compare medical cannabis to the standard therapy of fibromyalgia, to establish the proper place of cannabis in fibromyalgia therapeutic arsenal.

## Figures and Tables

**Figure 1 jcm-08-00807-f001:**
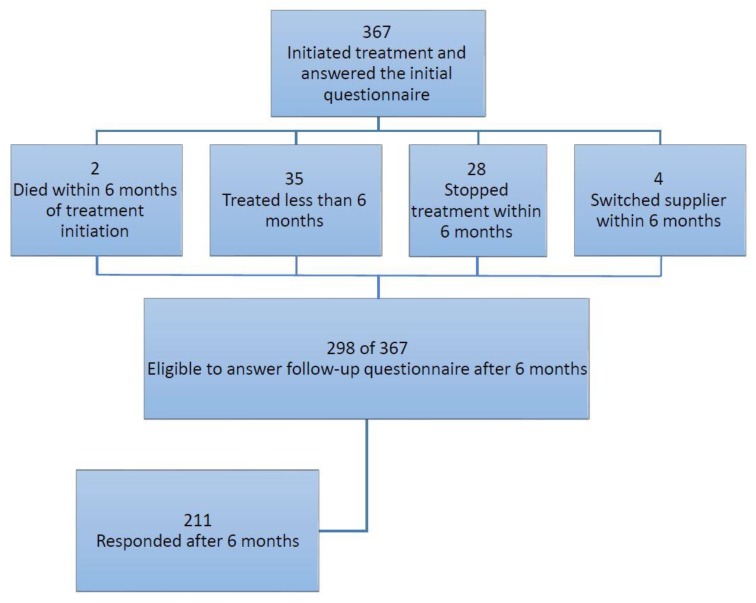
Flow chart of the study population.

**Figure 2 jcm-08-00807-f002:**
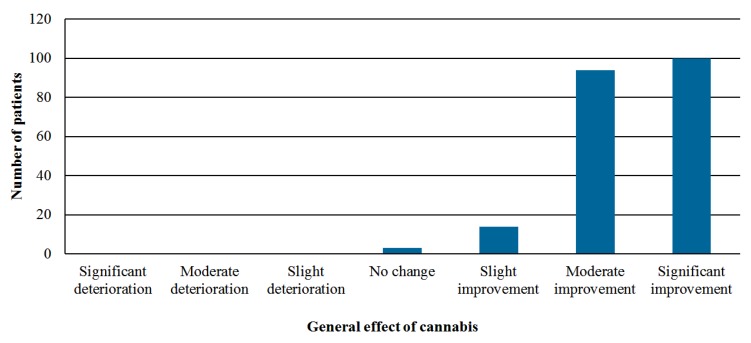
Perception of the general effect of cannabis on the patient’s condition after six months of treatment.

**Figure 3 jcm-08-00807-f003:**
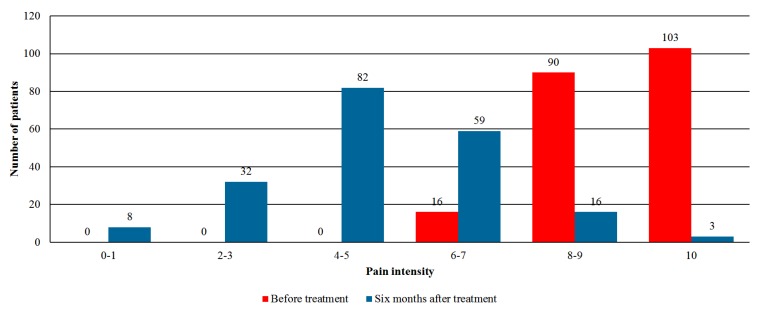
Assessment of the pain intensity on the 0–10 scale before and after six months of cannabis therapy (*p* < 0.001).

**Figure 4 jcm-08-00807-f004:**
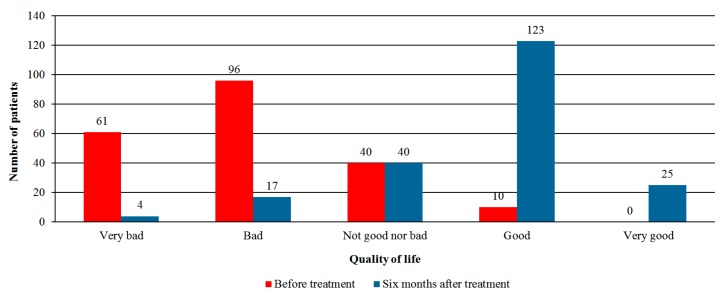
Quality of life prior and six months after the initiation of cannabis treatment (*p* < 0.001).

**Table 1 jcm-08-00807-t001:** Baseline characteristics of the patient population.

Variable	Number of Patients (*N* = 367)
Age (years), mean ± SD	52.9 (15.1)
Age groups, *n* (%)	
40 years and below	75 (20.4)
40–60 years	181 (49.3)
60 years and above	111 (30.2)
Females, *n* (%)	301 (82.0)
BMI (kg/m^2^), mean ± SD	28.6 (18.2)
Work status: works regularly	59 (16.1)
Part-time work	53 (14.4)
Unemployed/retired	233 (63.4)
Other	22 (5.9)
Driving a car, *n* (%)	231 (62.9)
Approved monthly dosage of cannabis ≤ 20 g, *n* (%)	328 (89.4%)
Approved route of administration, *n* (%)	Oil 74 (20.2) Inflorescence 247 (67.3) Oil + Inflorescence 44 (12.0)
Previous experience with cannabis, *n* (%)	166 (45.2)
Cigarette smokers, *n* (%)	137 (37.3)
Number of regularly used medications, median (i.q range)	5.0 (3.0–8.0)
Number of regularly used medications for fibromyalgia, median (i.q range)	1.0 (1.0–2.0)
Treatment indication: primary fibromyalgia, *n* (%)	283 (77.1)
Cancer, *n* (%)	35 (9.5)
PTSD, *n* (%)	22 (6.0))
Other, *n* (%)	27 (7.4)
Years of chronic pain, median (i.q. range)	7.0 (3.0–13.0)
Type of pain: Daily, *n* (%)	320 (87.2)
Episodic, *n* (%)	47 (12.8)

BMI—body mass index, SD—standard deviation I.Q range—interquartile range, and PTSD—post traumatic stress disorder.

**Table 2 jcm-08-00807-t002:** Multivariate analysis for treatment response at six months.

Variable	*p* Value	Odds Ratio	95% Confidence Interval
Age > 60 years	0.01	0.34	0.16–0.72
Concerns about cannabis-prior to treatment initiation	0.01	0.36	0.16–0.80
Spasticity	0.03	2.26	1.08–4.72
Previous experience with cannabis	0.04	2.46	1.06–5.74

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
