# Peer review of "Safety and Efficacy of Medical Cannabis in Fibromyalgia"

_jcm, 2019, doi:10.3390/jcm8060807_

Reviewer 1 Report

I thank the authors and the editors for the opportunity to review this study on the timely topic of medical cannabis. The topic is of importance in a global climate of changing legal landscapes and a focus on the opioid epidemic. However, this study leaves me with several questions, particularly regarding the choice of an uncontrolled study design for a question about treatment efficacy. This strikes me as a suboptimal choice of study design.

What are the specific objectives (i.e. a primary objective and secondary objectives if appropriate)? There is one sentence about the overall aim but it is not specific enough to be an objective.

There is no mention of the study design anywhere in the manuscript. It appears to be a case series (i.e. uncontrolled single group design). If this is the case, why was this design chosen? Uncontrolled designs are biased and do not assess efficacy very well. How do we know that patients actually improved? They may have improved on their own without the cannabis. It also isn't a blinded design so there could be expectation bias involved. If an RCT is not feasible, another design with an appropriately-selected control group would have been better than an uncontrolled design.

Why was loss to follow-up so high? It looks like the follow-up rate is only 57% from the study flow diagram. This is likely to bias results.

The authors conclude that medical cannabis is safe and effective, but they have not demonstrated this sufficiently for this strong of a conclusion. They acknowledge in their limitations section that this is uncontrolled observational data, but I still believe that the conclusion is too string for the data.

The manuscript could be improved by following a standard reporting guideline. E.g. STROBE for observational studies.

Funding and conflicts of interest should be disclosed for cannabis studies as they can be swayed by industry bias.

Reviewer 2 Report

This is a study reporting outcome of 367 persons with fibromyalgia (FM) and use of cannabis over a 6 month period. Any study assessing the benefits and risks of cannabis that is both well designed and is well conducted is greatly needed at this time.

Unfortunately there are so many concerns about this study at multiple levels that the outcomes reported by the authors have to be strongly questioned and challenged. I will outline just a few areas of great concern.

Major concerns

Two of the four authors report either been an employee of Ticum-Olam (the cannabis producer) or being a paid member of the scientific advisory board of Ticum-Olam.

Patients:

What is the denominator of all patients invited to participate? This study identified 367 FM patients. It is not stated how patients were recruited to participate, what numbers of those with FM accessing cannabis from Ticum-Olam declined to participate

Although this study had ethics approval, it is not stated that patients signed informed consent. And if this did occur, where was this done, ie by the rheumatologist, or at the time that the patient approached the cannabis provider Ticum-Olam Ltd.

It is not stated how patients were referred for cannabis treatment. Who referred patients, although it is stated there was a rheumatologist diagnosis of FM, it is not stated whether they were referred by the rheumatologist

Patient characteristics are completely outside the paradigm for FM…..a median pain score of 9/10 makes this study exceedingly suspect…this does never occur in either usual clinical practice, or even in studies reported from tertiary care centres.

23% of patients had FM as a secondary condition..including cancer, PTSD

Methodology and outcome measures:

Choosing of cannabis product and dosing. It is stated that “a nurse educates the patient and advises on selecting the cannabis strain and treatment dose according to titration protocol”. Nothing further is clarified. How does the nurse decide what to advise?

There is no prespecified primary or secondary outcome measure stated…rather a collection of various quasi measurements obtained over the phone.

Outcome measures are not validated usual outcomes and some are exceedingly puzzling, eg I fail to understand how vomiting, drop in sugar, cough, spasticity, confusion and disorientation are considered symptoms of FM, and change in these symptoms was recorded at 6 months. The authors boldly show impressive p values for this panoply of strange symptoms at 6 months in Supp Table 4.

The study product:

Although it is stated that the nurse could advise from 16 products, sup Table 2 identifies only 14 products, of which 9/14 have the same THC and CBD content!!!It is also stated in methods that patients initiated treatment with “a drop of 3% THC rich cannabis”…..but there is no 3% THC in Table 2.

I fail to understand how it was determined how “ the final dosage depended on the primary indication for cannabis use, age, medical background, parallel use of other analgesics regimens and previous exposure to cannabis”….particularly as it seems that patients were left to their own means without regular follow up

There is no statement of how compliance was assessed….there is no recording of study drug use on a monthly basis; how often did patients receive the study drug, there is no record of compliance such as a urine drug screen for cannabis.

Follow up:

Although it is stated that there was 70.8% fu, this is not true according to numbers…follow up was only 57%

Numbers are hard to follow, at times values are for 211 patients, and at other times for 239 patients.

The ethics of sound medical practice is questioned. These patients were provided with cannabis, seemingly after a face to face interview with a nurse, but follow up was by a telephone call at 1 month and 6 months. This is in the context of using a psychoactive substance in persons using a median of 5 other drugs, with 118 reported to be using benzodiazepines and 126 using opioid analgesics.

 Author Response

Please see the enclosed file

Reviewer 3 Report

Manuscript review

Safety and Efficacy of Medical Cannabis in FM

Thankyou for the opportunity to review the manuscript. The authors have used the novel context of the Israel Medical Cannabis Agency who approve use of prescribed medical cannabis to examine the safety and efficacy of use in people with Fibromyalgia (FM). The manuscript is well written and easy to follow, but several concerns remain and are detailed below.

Abstract:

Though preclinical data show promise and there is strong consumer demand, the opening sentence of the abstract needs to be moderated, reworded, or referenced. The only systematic review that examines RCT outcomes for the efficacy, tolerability and safety of cannabinoids in chronic pain (Fitzcharles et al 2016) Schmerz 30: 46), concludes that “…there is insufficient evidence for recommendation for any cannabinoid preparations for symptoms management in patients with chronic pain…” also see Campbell, G(2018) Lancet Public Health 3:e341-e350 prospective study in people with chronic non-cancer pain “..we found no evidence that cannabis use improved patient outcomes.”

Background

The authors have established the rationale for the study – FM is a big problem and current therapies are not marvellously effective so let’s examine cannabis which is safe and effective the way it is prescribed in Israel. Well done.

Experimental section  

The design did not incorporate any measure of another symptom commonly associated with FM – fibro “fog” or difficulty with higher order cognitive processing and memory. Given concerns about cannabis sideeffects related to cognitive function could the authors please present the reasons why they didn’t examine this aspect? As it would be difficult to redress this aspect, it would seem like a necessary addition to the limitations section.

The cannabis dose was titrated for therapeutic effect, but how therapeutic effect was defined was not described. Could the authors please describe the criteria by which therapeutic effect was defined?

Participants received one of 14 different strains of cannabis. How was the strain of cannabis determined?

Analysis was by telephone interview at 1 and 6 months with a 70.8% response rate at 6 months. Pain intensity was reported to be assessed on a VAS – did the participants have a visual representation of the VAS in from of them at the time they were speaking on the phone? If not would the authors consider that they actually used a numeric rating scale (NRS) to record pain intensity?

Results

Given the large number of strains and the potential for many different doses at each strain, different formulations and highly variable pharmacokinetics would the design have been improved by limiting inclusion criteria to FM (an already heterogenous disorder) as the primary pain related indication. Could the authors comment on the potential agents responsible for therapeutic effects in if the primary indication for cannabis therapy was cancer or PTSD? It seems difficult to unravel the underlying mechanisms that might be addressed by the cannabis therapy which is the case overall, but varying the inclusion criteria may compound this?

On average the mean THC dose was higher than the CBD dose, could the authors explain what the implications of this are – what are the expected therapeutic outcomes of this ratio of components. Would a different ratio suit another type of disorder better? Is there an index of pharmacological effect that is considered to indicate successful/efficacious therapy?

Figure 3 shows a normalisation of pain intensity score distribution following treatment (baselines assessment showed a skewed distribution). Have the authors speculated on why normalisation of the distribution of scores occurred? Is there any capacity to comment on the clinical relevance of the change in scores?

Discussion

The opening paragraph suggests that there was substantial improvement in pain intensity – could the authors please define ‘substantial’.

There is also a claim that there was significant improvement in QOL after 6 months, but a more accurate report would suggest that only certain features of QOL were improved significantly which in fact 6 activities of daily living deteriorated significantly. To report both these findings would make a difference to the quick reader who on current scanning would suggest this study provides evidence of a positive overall effect. Would the authors please give the report and wording a little more consideration in light of these comments?

The second paragraph gives the context into which this study fits, but does not integrate this study. Could the authors please relate their study and outcomes to the known literature presented in this paragraph? Perhaps this could be done by reorganising the contents of paragraph 2 and 3 to integrate what is perceived as a strength of this study against the outcomes of each of the 3 studies?

The observations about only a 30% decrease in pain intensity with current therapy could also apply to this study. As a reader we don’t know whether a baseline report of 10 came down to a 6 month followup report of 7 or 8 or in fact 2. It would be interesting to find out not just the group change scores, but also individual trajectories, certainly if you’d like to start unravelling the effects of different strains and doses?

Opioid use in FM is not effective (Borchers and Bershwin 2015 Clin Rev Allergy and Immunol 49: 100)  and it is surprising to see that so many were using opioids at baseline. A head to head comparison with cannabis therapy would be a questionable and irresponsible research proposal. Even more so with the current complexity of opioid overuse. Could the authors please remove this suggestion from the discussion?

The safety issue and side effects did not address the association between cognitive function and FM and cannabis. There is a place for further study into this aspect which may indeed compromise safe use of the drug in this particular population. I wonder whether the authors might like to add a future or follow up research suggestions section to the paper?

References

Please check the reference manager record for Arthritis care and research  and Cannabis and cannabinoid research which appear un abbreviated and without capitals and so in contrast to the other references in the list. The second P in Pain physician might need capitalisation as well.

Author Response

Please see the enclosed file

Round  2

Reviewer 1 Report

The authors have addressed my comments sufficiently. The authors admit that an RCT would be a better design, but I think this uncontrolled study can also add insight to the literature.

Author Response

We thank the reviewer for the opportunity to improve the manuscript and hope it would be suitable to be published in its current form.